# Biomarker Expression of Peri-Implantitis Lesions before and after Treatment: A Systematic Review

**DOI:** 10.3390/ijerph192114085

**Published:** 2022-10-28

**Authors:** Haniyeh Moaven, Annesi Giacaman, Víctor Beltrán, Ye Han Sam, Daniel Betancur, Giuseppe Mainas, Seyed Ali Tarjomani, Nikolaos Donos, Vanessa Sousa

**Affiliations:** 1Centre for Oral Clinical Research, Centre for Oral Immunobiology & Regenerative Medicine, Institute of Dentistry, Barts and The London School of Medicine and Dentistry, Queen Mary University London, London E1 2AD, UK; 2Center of Excellence in Translational Medicine, Faculty of Medicine, Universidad de la Frontera, Temuco 4780000, Chile; 3Clinical Investigation and Dental Innovation Center, Dental School & Center for Translational Medicine, Universidad de La Frontera, Temuco 4780000, Chile; 4Periodontology and Periodontal Medicine, Centre for Host-Microbiome Interactions, Faculty of Dentistry, Oral & Craniofacial Sciences, King’s College London, Guy’s and St Thomas’ NHS Foundation Trust, London SE1 9RT, UK; 5Discipline of Periodontology, Department of Surgical Stomatology, Faculty of Dentistry, Universidad de Concepción, Concepción 4030000, Chile

**Keywords:** biomarkers, peri-implantitis, inflammation, therapy, laser, amelogenin

## Abstract

The need to predict, diagnose and treat peri-implant diseases has never been greater. We present a systematic review of the literature on the changes in the expression of biomarkers in peri-implant crevicular fluid (PICF) before and after treatment of peri-implantitis. Bacterial composition, clinical and radiographic parameters, and systemic biomarkers before and after treatment are reported as secondary outcomes. A total of 17 studies were included. Treatment groups were non-surgical treatment or surgical treatment, either alone or with adjunctive therapy. Our findings show that non-surgical treatment alone does not influence biomarker levels or clinical outcomes. Both adjunctive photodynamic therapy and local minocycline application resulted in a reduction of interleukin (IL)-1β and IL-10 twelve months after treatment. Non-surgical treatments with adjunctive use of lasers or antimicrobials were more effective at improving the clinical outcomes in the short-term only. Access flap debridement led to matrix metalloproteinase (MMP)-8 and tumour necrosis factor-α reduction twelve months post-surgery. Surgical debridement with adjunctive antimicrobials achieved a decrease in MMP-8 at three months. Adjunctive use of Emdogain^™^ (EMD) was associated with a reduction in 40 PICF proteins compared to access flap surgery alone. Surgical interventions were more effective at reducing probing pocket depth and bleeding on probing both in the short- and long-term. Surgical treatment in combination with EMD was found to be more effective in resolving inflammation up to twelve months.

## 1. Introduction

Healthy peri-implant tissues are characterized by the absence of clinical evidence of inflammation (erythema, bleeding on probing (BOP), swelling and suppuration), and the absence of additional bone loss following initial healing [1]. Peri-implantitis is “a plaque-associated pathological condition occurring in tissues around dental implants” [1] which demonstrates peri-implant mucosal inflammation, supporting bone loss, and increasing probing depth. The prevalence of peri-implantitis is reported to be 24% (range 1–47%) [2] and is expected to increase annually [3].

The sensitivity of clinical and radiographic investigations of peri-implant inflammatory diseases are low, and detection only occurs once the tissue damage has already taken place. Current research avenues are directed towards supplementary, non-invasive techniques, where early detection of pathological biological activities is possible. Furthermore, innovations in implant dentistry have been documented in the quest to reduce morbidity, biological complications, and surgical times [4,5].

The diagnosis of peri-implant disease is mostly made using imaging and clinical parameters like the clinical attachment level and BOP. Investigating associations between certain biomarkers with health and/or disease can provide clinicians with better tools to reach an accurate diagnosis [6] and monitor disease activity [6]. Saliva, serum, sub-gingival plaque and gingival crevicular fluid (GCF) are all sources of biomarkers for periodontal disease [7]. Similar to GCF, peri-implant crevicular fluid (PICF) contains cytokines and other biomarkers, microorganisms, and host cells reflecting physiological and pathological host-microflora interactions [7]. The biomarkers present in the GCF and PICF are related to the inflammatory response, and they are currently being studied as key biomarkers to determine the health or disease status of dental implants [6].

Monitoring the progression from peri-implant mucositis to peri-implantitis is challenging, and tracking changes in these inflammatory biomarkers may help to detect the onset of early disease and to track the disease activity. Researchers have previously focused on biomarkers such as receptor activator of nuclear factor kappa-Β ligand (RANKL), osteoprotegerin (OPG) and associated biomarkers, as they have shown promising results in distinguishing between health and disease status [6]. Furthermore, when peri-implantitis is already present, the destruction of the peri-implant bone and soft tissues has a high variability between patients. In these circumstances, biomarkers could play a vital role in establishing a new minimally invasive approach. 

Interleukin (IL)-23 is an inflammatory cytokine and is an IL-12 family member. It regulates the maintenance and expansion of T-helper 17 (Th17) cells. This cytokine is mainly secreted by activated monocytes, macrophages, and dendritic cells. IL-23 stimulates Th-17 cells to produce IL-17, which subsequently stimulates RANKL. RANKL proteins are involved in regulating the immune response and alveolar bone metabolism. 

Chemokine ligand-20 (CCL20), also known as liver activation regulated chemokine (LARC) or macrophage inflammatory protein-3 (MIP-3α), is a strong chemotactic agent for B- and T-lymphocytes and weakly attracts neutrophils. By attracting and activating B- and T-lymphocyte, the main producers of RANKL, a cascade is triggered that results in modulation of the inflammatory response and promotion of osteoclast differentiation and activation [8].

B-cell activating factor (BAFF, also known as BLyS) and a proliferation-inducing ligand (APRIL) are TNF ligands that play an important role in B-lymphocyte differentiation, maturation and survival. B-cell chemotaxis and proliferation in the periodontal tissues leads to an increased chronic inflammatory response and facilitates alveolar bone destruction, as these cells express RANKL.

Based on the limited available evidence on the success of different treatment modalities for peri-implantitis, the aim of this systematic review is to explore biomarker profile before and after treatment and its translation into clinical findings in the short and long term.

## 2. Materials and Methods

### 2.1. Study Design

The current systematic review followed the Preferred Reporting Items for Systematic review and Meta Analyses (PRISMA) statements [9] to gather available evidence on the focus question: What are the differences in the PICF human biomarker expression before and after the treatment of peri-implantitis?

The protocol for this systematic review was registered in the Open Science Framework registration ID JN97F (Figure 1). 

Specific descriptors were organized according to the PICO strategy (Patient, Intervention, Comparison and Outcome) as follows: A.Population: Studies including:
Patients aged ≥18 years;Diagnosis of peri-implantitis;Reporting on baseline (untreated) and post-treatment PICF biomarker concentration.
B.Intervention: Non-surgical and surgical treatment of peri-implantitis.C.Comparison: Participants with healthy peri-implant tissues (diagnosis of peri-implant health).D.Outcomes:
Primary Outcome: Reported biomarker concentrations in PICF before and after treatment of peri-implantitis.Secondary Outcomes: Report of systemic biomarkers, and biomarker concentration in saliva and/or other tissues as reported by the authors (in participants diagnosed with peri-implant health, peri-implant mucositis, and peri-implantitis), and PICF volume. Bacterial (microbiome) reports of composition of peri-implantitis lesions before and after treatment, and peri-implant health. Clinical parameter changes, e.g., Probing Pocket Depth (PD), Clinical Attachment Loss (CAL), recession, BOP, suppuration, Plaque Index (PI), phenotype of keratinised tissue, vertical/horizontal components of keratinised tissue. Radiographic parameters, e.g., bone levels/loss.

### 2.2. Search Strategy

The search strategy was constructed using PICO terms and electronic search in PubMed, MEDLINE, Embase, Google Scholar, Scopus, Web of Science, ProQuest and WHO International Clinical Trials Registry Platform. A search for grey literature was done in Open Grey and at ClinicalTrials.gov. Manual search through bibliographies of included studies was also performed. The search was aimed at evidence identified up to September 2022.

A.Eligibility Criteria

The inclusion criteria were as follows: randomized controlled clinical trials (RCTs); prospective case-control studies; prospective cohort studies; prospective case series; retrospective, prospective and cross-sectional observational studies; studies including direct comparison of biomarker concentrations in peri-implantitis before and after treatment; adult patients.

The exclusion criteria were as follows: systematic reviews and/or meta-analyses; studies lacking a baseline comparison prior to commencing treatment for peri-implantitis; extra-oral implants, zygomatic implants and implants related to orthodontics; pre-clinical, in vivo and in vitro studies.

B.Selection of Studies

Results screening was performed by two independent reviewers (HM and AG). Disagreements were discussed among reviewers at each stage, and the disputed studies were discussed with a third reviewer (VS). Firstly, duplicates were removed; then, title screening was performed to eliminate irrelevant studies. Next, abstract, and full text screening was performed, and studies not meeting the inclusion criteria were excluded. The level of agreement between all the reviewers were assessed using Kappa statistics at each stage (1.00) [10].

### 2.3. Data Extraction

The data extraction tables were created based on the PICO question and included both qualitative and quantitative data. The following study characteristics were recorded: author, year of publication, study design, number of study arms, sample size calculation, number of participants, gender of participants, gender distribution, mean age, source of recruitment, proportion of participants with history of periodontitis (in total as well as in the test and control groups), primary outcomes, secondary outcomes, additional outcomes, peri-implantitis definition, implant position, type of prosthesis, type of procedure and clinical details, and laboratory technique.

### 2.4. Quality Assessment of Included Studies

The Quality Assessment of Diagnostic Accuracy Studies-2 (QUADAS-2) tool was used to assess the quality and risk of bias for all papers included in this systematic review [11]. This tool is recommended by the Agency for Healthcare Research and Quality, Cochrane Collaboration [12] and the UK National Institute for Health and Clinical Excellence for systematic reviews focusing on diagnostics accuracy studies [11].

### 2.5. Data Synthesis

A descriptive analysis of the findings was used to evaluate the data. Qualitative data synthesis was performed resulting in a narrative report. Qualitative analyses were performed by summarizing the type of intervention, the outcomes, an indication of the study’s findings, presented in evidence tables, and a resultant narrative report. 

## 3. Results

### 3.1. Study Selection

Total search results numbered 24,431, of which 12,236 were duplicates. Title screening resulted in 280 articles for further detailed evaluation. Eventually, 188 studies met the criteria for full text screening (Figure 1) of which 17 studies were included for data extraction and qualitative synthesis. All screenings were carried out by two independent reviewers.

### 3.2. Study and Patient Characteristics

The characteristics of the included studies are summarized in Table 1. All 17 included studies [13,14,15,16,17,18,19,20,21,22,23,24,25,26,27,28,29] were published in English. Sample sizes ranged between 10 and 48 patients. There were nine RCTs [13,17,18,20,22,23,25,27,29], six prospective case-control studies [14,16,19,21,24,28] and two case series [15,26].

The inclusion criteria for peri-implantitis diagnosis in all but one study was PD > 4 mm with BOP and/or suppuration on probing and radiographic evidence of marginal bone loss after prosthesis delivery. In one study, no criteria for peri-implantitis were reported. Ref. [19] Details of case definition per study are summarized in Table 1.

The type of procedure included in the test groups for the treatment of peri-implantitis consisted of non-surgical mechanical debridement alone [16,21], non-surgical therapy with adjunctive use of lasers [13,23,26,28], non-surgical therapy with adjunctive use of antimicrobials, [13,18,20,22,24] access flap surgery alone [15,16,19], access flap with adjunctive antimicrobials [14] and access flap with EMD (Enamel Matrix Derivatives) [17]. In two studies, following induction of peri-implant mucositis, different formulations of toothpaste were assessed in terms of efficacy of the resolution of inflammation [25,27].

Biomarkers studied were as follow: IL-1α [23], IL-1b [13,18,20,21,22,23], IL-1RA [20], IL-4 [16,20], IL-6 [20,21,23], IL-8 [13,20,23], IL-10 [13,16], IL-12 [16], IL-17A [20], Matrix Metalloproteinase (MMP)-1 [13,23], Macrophage Inflammatory Protein (MIP)-1α [14,26], Tumour Necrosis Factor (TNF)-α [15,16,20], RANKL [16,21,25], OPG [16,21,25,29], CCL5 [20], Interferon (IFN)- γ [20,21], MMP-1 [13,23], MMP- 3 [23], MMP-8 [29], MMP-9 [23], MMP-13 [23], MMP- 96 [25], Tissue Growth Factor (TGF)-β7 [25], C-Reactive Protein (CRP) [23], Granulocyte Macrophage Colony-Stimulating Factor (GM- CSF) [20], Myeloperoxidase (MPO) [24], Alkaline Phosphatase (ALP) [24], IP-10 [26], Platelet-Derived Growth Factor (PDGF)-BB [26], Vascular Endothelial Growth Factor (VEGF) [26], osteocalcin [25,29], leptin [29], osteopontin [25,29], parathyroid hormone [29], adiponectin [29], insulin [29], total protein content [17], MCP-1/CCL2 [21], MIP-1α/CCL3 [21], and Soluble RANK Ligand (sRANKL) [21]. 

Only one study reported proteomics data of the collected PICF samples [17].

There was heterogeneity amongst studies on PICF sample analysis in terms of the techniques employed for biomarker assessment. Biomarker assessment was carried out using ELISA [13,15,16,19,22,28,29], Luminex [14,21,25,26,27], Driscol [18], Bio-Plex [20], multiplex suspension array [23] and spectrophotometry [24].

The follow-up period in the included studies ranged between 3 weeks and 12 months. 

### 3.3. Primary Outcome Results

The results of the effect of different peri-implantitis treatment modalities on biomarker concentrations are summarized in the Appendix A.

#### 3.3.1. Non-Surgical Treatment

In two studies, non-surgical treatment of peri-implant mucositis and peri-implantitis reported minimal effect on inflammatory biomarker levels [16,21]. Duarte et al. [16] performed non-surgical treatment using abrasive air powder and resin curette and Hentenaar et al. [21] used Airflow Master Piezon (EMS).

Amongst investigated biomarkers (IL-4, IL-10, and IL-12, TNF-a, RANKL, and OPG in PICF), Duarte et al. [16] showed that although all biomarkers concentration reduced after treatment, results were only statistically significant for TNF-a at 3 months after treatment [16]. From the investigated biomarkers in Hentenaar et al. [21] (IL-1β, IL-6, TNF-α, MCP-1/CCL2, MIP-1α/CCL3, IFN-γ, MMP-8, sRANKL, OPG and G-CSF), the authors indicated that non-surgical treatment of peri-implantitis did not have clear benefits in terms of reduction of inflammation (as expressed by the specific biomarkers) [21].

Adjunctive use of laser to non-surgical treatment was investigated in four of the studies [13,23,28,30]. In an RCT comparing photodynamic therapy (PDT) to local drug delivery (LDD) of minocycline, Bassetti et al. [13] reported a significant reduction of IL-1b and IL-10 in both groups at 12 months post treatment. However, for the rest of the investigated biomarkers (IL- 8, MMP-1 and MMP-8), the reduction was not statistically significant in either of the groups [13]. Komatsu et al. [23] in a RCT, compared the effectiveness of Erbium-Doped: Yttrium, Aluminum, and Garnet (Er:YAG) laser vs. locally delivered minocycline hydrochloride ointment in treatment of peri-implant disease. Comparison between two groups showed a statistically significant reduction of MMP-9 levels in the Er:YAG group [23]. Renvert et al. [26] in a case series, compared biomarker levels after treatment of peri-implantitis with PerioFlow vs. Er:YAG laser. The results showed that in clinically stable cases, levels of IL-1b, VEGF and IL-6 were lower in both groups [26]. Thierbach et al. [28] studied the level of MMP-8 in smokers and non-smokers with peri-implantitis undergoing non-surgical (antimicrobial PDT) and surgical treatment (access flap and laser irradiation 4 months after non-surgical phase). The results showed no significant change of MMP-8 level at baseline and post treatment in both groups [28]. 

Adjunctive use of Azithromycin (AZM) as to non-surgical treatment of peri-implantitis has been investigated in two studies [18,22]. Gershenfeld et al. [18] in an RCT, performed mechanical debridement for all patients; the test group received AZM 500mg once daily for three days, and controls received a placebo. Biomarker (IL-1b) levels were reduced in both groups without a statistically significant difference between test and control groups [18]. Kalos [22] in a pilot study compared non-surgical treatment with and without adjunctive AZM and reported similar results to Gershenfeld et al. [18]. 

Malik et al. [24] conducted a case-control study and evaluated the levels of ALP and MPO by comparing results observed in healthy peri-implant mucosa with peri-implant sites which received non-surgical anti-infective therapy (supra and subgingival scaling supplemented by local irrigation with 0.2% chlorhexidine mouthwash and post-operative prescription of chlorhexidine gel for 4 weeks). The results showed the peri-implantitis group had statistically significant higher MPO and ALP at baseline and 3 months post treatment compared to healthy sites (*p* < 0.001). Non-surgical treatment proposed in the study did not result in a statistically significant reduction of mentioned biomarkers [24].

One study investigated the effect of adjunctive probiotics to non-surgical treatment of peri-implant mucositis [20]. In this study, after initial mechanical debridement and oral hygiene instructions (OHI), the patients received a topical oil application (active or placebo) followed by twice-daily intake of lozenges (active or placebo) for 3 months. The active products contained a mix of two strains of *Lactobacillus reuteri* (probiotics). During the final assessment, 12 weeks after treatment, there were reduced levels of the pro-inflammatory cytokines during the intervention period in both groups compared with baseline, but there was no statistically significant difference between the two groups [20].

Adjunctive use of Triclosan in dentifrices for treatment of experimental peri-implant mucositis has been investigated in two studies [25,27]. In an experimental study, Pimentel et al. [25] investigated the effects of triclosan-containing fluoride toothpaste on the level of biomarkers and compared it with non-triclosan-containing fluoride toothpaste on patients who were instructed to refrain from performing oral hygiene measures for 3 weeks. Results showed no intra- or inter-group differences for IFN-γ, IL10, IL-1β, IL8, IL-17, IL-6, TNF-α, MMP-2, MMP-9, TGF-β, OC, OPN, Crosslinked Telopeptide of Type I Collagen (ICTP), OPG and RANKL (*p* > 0.05). RANKL/OPG ratio was significantly higher in fluoride toothpaste-treated sites when compared to triclosan/fluoride-treated sites at the end of period without mechanical tooth brushing, on the 21st day (*p* = 0.041) [25]. Ribeiro et al. [27] with similar study design showed that level of IL-10 was reduced and IL-1β concentrations were increased at three weeks when compared with baseline only in placebo-treated sites (*p* < 0.05). OPG levels significantly increased from week two to week three only in sites treated with triclosan (*p* < 0.05) [27].

#### 3.3.2. Surgical Treatment Outcomes

Three studies investigated the effect of access flap debridement on biomarker level after treatment [15,16,19]. Granfeldt et al. [19] in a case-control study investigated MMP-8 levels after treatment of peri-implantitis with access flap alone vs. access flap with addition of porous titanium particles. Results showed a statistically significant reduction in MMP-8 levels after 12 months compared to baseline. However, no statistically significant reduction in MMP-8 between the two groups was shown [19]. De Mendonca et al. [15] in a case-series investigated the level of TNF-α in PICF of implants undergoing access flap surgery and debridement with abrasive sodium carbonate air-powder and resin curettes. Results at 12-month post intervention showed statistically significant reduction in the PICF biomarker level [15].

The adjunctive use of lasers during access flap debridement in the treatment of peri-implantitis was investigated in a study by Thierbach et al. [28]. There were no statistically significant differences after treatment in smokers and non-smokers with peri-implantitis [28].

Adjunctive use of antibiotics with access flap debridement was investigated in a case-control study by Bhavsar et al. [14]. The results showed significant reduction of MMP-8 post treatment in the test group but still higher than the control group. Other investigated biomarkers (IL-1β and MIP-1α) showed no significant reduction in concentration after treatment, when compared to baseline [14].

Wohlfahrt et al. [29] investigated the efficacy of access flap debridement and surface decontamination with titanium curettes and 24% Ethylenediaminetetraacetic Acid (EDTA) gel vs. additional insertion of porous titanium granules in treatment of peri-implantitis. No difference in bone marker levels nor other biomarkers were found between the test and control group either at baseline or at 12 months [29].

Recently, Esberg et al. [17] investigated adjunctive use of EMD to access flap and its effect on the proteomic profiles. Results showed that EMD treatment is significantly associated with a decreased prevalence of 40 PICF proteins [17].

### 3.4. Secondary Outcomes

A summary of the results of the effect of different treatment modalities of peri-implantitis on secondary outcomes is presented in the Appendix A.

#### 3.4.1. Non-Surgical Treatment Alone

A.Probing Pocket Depth:

Duarte et al. [16] reported a statistically significant reduction in PD after treatment, whereas Hentenaar et al. [21], following non-surgical therapy, showed no statistically significant PD reduction [16,21].

B.Radiographic Bone Level:

Hentenaar et al. [21] reported that, following treatment, the marginal bone level around implants was reduced. However, this bone loss was not statistically significant when evaluated both clinically and radiographically [21].

C.Bleeding on Probing:

Duarte et al. [16] reported statistically significant reduction in BOP three months after treatment, whereas Hentenaar et al. [21] showed that BOP reduction, following non-surgical therapy, was not significant [16,21].

D.Suppuration:

Duarte et al. [16] reported a statistically significant reduction in suppuration three months after treatment, whereas Hentenaar et al. [21] showed that suppuration, following non-surgical therapy, was slightly increased at implant site [16,21].

E.Plaque Index:

Both Duarte et al. [16] and Hentenaar et al. [21] showed reduction in PI following non-surgical treatment [16,21].

F.Peri-Implant Crevicular Fluid Volume:

Hentenaar et al. [21] reported a reduction of mean PICF volume after non-surgical treatment. However, PICF volume was still higher than healthy peri-implant sites [21].

No studies reported information concerning clinical attachment levels, recession or the microbiome.

#### 3.4.2. Non-Surgical Treatment with Adjunctive Use of Laser

A.Probing Pocket Depth:

Bassetti et al. [13], reported a statistically significant reduction in PD after treatment with Adjunctive Photodynamic Therapy (a-PDT) up to 9 months, but no statistically significant difference at 12 months. The Adjunctive Local Drug Delivery (LDD) group showed a statistically significant reduction in PD up to 12 months post treatment [13]. 

Komatsu et al. [23] reported a statistically significant reduction in PD after treatment up to 3 months in the group treated with Er:YAG laser, whereas PD was not significantly reduced in the group treated with local delivery of minocycline [23]. Renvert et al. [26] reported clinically stable reduction in PD at 6 months after treatment of peri-implantitis by using PerioFlow or Er:YAG laser [26].

B.Clinical Attachment Levels:

Bassetti et al. [13] reported no statistically significant differences in clinical attachment levels between a-PDT and LDD groups at any time point after treatment [13].

C.Recession:

Bassetti et al. [13] reported a statistically significant reduction in recession in the a-PDT group up to 9 months. The LDD group showed a statistically significant reduction in recession up to 6 months [13].

D.Bleeding on Probing:

Bassetti et al. [13] reported a statistically significant reduction in BOP% up to 12 months in both groups (a-PDT and LDD) and similar results were reported by Komatsu et al. [23] at 3 months. Renvert et al. [26] reported a reduction in overall BOP% in both groups, by using either PerioFlow or Er:YAG [26]. 

E.Suppuration:

Duarte et al. [16] reported a statistically significant reduction in suppuration at 3 months after treatment while Hentenaar et al. [21] showed that suppuration following non-surgical therapy was slightly increased at implant sites [16,21]. Renvert et al. [26] reported reduction in overall suppuration in both groups, by using either PerioFlow or Er:YAG [26]. 

F.Plaque Index:

Bassetti et al. [13] reported a statistically significant reduction in PI up to 12 months in both groups (a-PDT and LDD).

G.Microbiome Analysis:

Bassetti et al. [13] reported that except for *Campylobacter rectus (C. rectus**)* at baseline, the counts in the sub-gingival biofilm were not statistically significantly different between test and control groups at any time point. *Porphyromonas gingivalis* (*P. gingivalis*) and *Tannerella forsythia (T. forsythia)* significantly reduced from baseline to 6 months in a-PDT group and to 12 months in LDD group. At baseline, the most frequently identified species in the sub-mucosal biofilm were *Capnocytophaga gingivalis (C. gingivalis)*, *Fusobacterium nucleatum (F. nucleatum)*, *Parvimonas micra (P. micra)* and *T. forsythia*. 

Komatsu et al. [23] reported no statistically significant differences for both gram-positive and gram-negative bacteria in laser group at any time point whereas, in test group, with patients treated with local minocycline, a statistically significant decrease of all bacterial groups at 3 months was observed [23].

#### 3.4.3. Non-Surgical Treatment with Use of Adjunctive Antimicrobials 

A.Probing Pocket Depth:

Gershenfeld et al. [18] reported that a single dose of AZM as an adjunct to non-surgical treatment leads to greater PD reduction compared to placebo [18]. Bassetti et al. [13] compared a-PDT with LDD and reported that the LDD group showed a statistically significant reduction in PD up to 12 months following treatment [13]. Hallstrom et al. [20] reported a PD reduction after adjunctive use of probiotics in non-surgical treatment of peri-implant mucositis, without a significant difference between two groups [20]. Malik et al. [24] reported a PD reduction after adjunctive use of chlorhexidine with subgingival irrigation in non-surgical treatment of peri-implant mucositis, but there was no significant difference compared to the baseline [24].

B.Clinical Attachment Levels:

Bassetti et al. [13] reported no statistically significant difference in clinical attachment levels between a-PDT and LDD groups at any time point after treatment [13].

C.Recession:

Gershenfeld et al. [18] reported that single a dose of AZM as an adjunct to non-surgical treatment leads to slight improvement, whereas in the placebo group a slight increased recession was recorded [18]. Bassetti et al. [13] reported that recession in LDD group was statistically significantly reduced up to 6 months after treatment [13].

D.Radiographic Bone Levels:

Gershenfeld et al. [18] reported no statistically significant difference in radiographic bone level change in test (non-surgical with AZM) and control (non-surgical with placebo) groups.

E.Bleeding on Probing:

Gershenfeld et al. [18] reported that a single dose of AZM as an adjunct to non-surgical treatment leads to greater pocket depth reduction compared to placebo [18].

Bassetti et al. [13] reported statistically significant post treatment reduction in BOP% up to 12 months in both studied groups (a-PDT and LDD) [13].

Hallstrom et al. [20] reported a 5% reduction in BOP in both groups (post adjunctive probiotics and placebo to non-surgical treatment) after treatment of peri-implant mucositis [20]. Malik et al. [24] reported BOP reduction after use of chlorhexidine subgingival irrigation in non-surgical treatment of peri-implant mucositis, without any significant difference compared to baseline [24].

F.Suppuration:

Hallstrom et al. [20] reported reduced suppuration in both groups (adjunctive probiotics and placebo to non-surgical treatment) after treatment of peri-implant mucositis [20].

G.Plaque Index:

Bassetti et al. [13] reported statistically significant reduction in PI up to 12 months in both groups (a-PDT and LDD [13]. Hallstrom et al. [20] reported plaque index reduced in both groups (adjunctive probiotics and placebo to non-surgical treatment) regarding the treatment of peri-implant mucositis [20].

H.Microbiome Analysis:

Gershenfeld et al. [18] reported that orange complex bacteria had the highest frequency between two groups (non-surgical treatment + adjunctive AZM vs. Non-Surgical Periodontal Treatment (NSPT) + placebo) followed by purple, green and yellow complex. Interestingly, red complex bacteria had the least positive frequency at all time points [18]. Kalos et al. [22] with a similar study design to the previous study reported no statistically significant difference between treatment groups in aerobic and anaerobic bacterial count [22]. Bassetti et al. [13] reported that at baseline, the most frequently identified species in the sub-mucosal biofilm were *P. gingivalis*, *Fusobacterium nucleatum (F. nucleatum)*, *P. micra* and *T. forsythia*. There was no statistically significant difference over time between groups. *P. gingivalis* and *T. forsythia* were significantly reduced from baseline to 6 months in a-PDT group and to 12 months in LDD group [13]. 

Hallstrom et al. [20] reported no statistically significant difference between two groups (adjunctive probiotics and placebo to non-surgical treatment) in treatment peri-implant mucositis. The most prevalent strains were *F. nucleatum*, *P. micra*, *Prevotella intermedia* and *Prevotella nigrescens* [20].

#### 3.4.4. Access Flap Surgery Alone 

A.Probing Pocket Depth:

De Mendonga et al. [15] reported access flap surgery and use of abrasive air powder and resin curette significantly reduced PD at 12 months post treatment [15]. Duarte et al. [16] with similar study design, reported significantly reduced PD at 3 months post treatment both in peri-implant mucositis and peri-implantitis groups compared to baseline [16].

B.Clinical Attachment Levels:

De Mendonga et al. [15] reported access flap surgery and use of abrasive air powder and resin curette led to no statistically significant difference in clinical attachment levels compared to baseline at 12 months [15].

C.Bleeding on Probing:

De Mendonga et al. [15] reported access flap surgery and use of abrasive air powder and resin curette significantly reduced BOP% at 12 months post treatment [15]. 

Duarte et al. [16] with similar study design reported significantly reduced BOP% at 3 months post treatment in the peri-implantitis group compared to baseline [16].

D.Suppuration:

De Mendonga et al. [15] reported access flap surgery and use of abrasive air powder and resin curette significantly reduced suppuration percentage at 12 months post treatment [15].

E.Plaque Index:

Duarte et al. [16] reported significantly reduced plaque index at 3 months post treatment in both peri-implant mucositis and peri-implantitis groups compared to baseline [16]. 

#### 3.4.5. Access Flap Surgery with the Use of Adjunctive Antimicrobials 

Bhavsar et al. [14] investigated adjunctive use of prophylactic antibiotics to access flap debridement in a case-control study. Adjunctive antibiotics resulted in significant reduction of PD, BOP% and suppuration compared to baseline [14].

### 3.5. Risk of Bias within Studies Results 

The risk of bias and applicability judgements of the included studies using QUADAS-2 tool [11] are reported in Appendix A, and excluded studies following eligibility criteria assessment are shown in Appendix A. 

A high risk of bias for participant selection was shown in nine (44%) studies [14,15,16,18,19,24,26,28]. Low risk of bias in participant selection was reported in four studies [13,20,23,29], and the remaining six studies had unclear risk of bias [17,18,22,24,25,27]. The risk of bias for the index test was high for all the studies except two [15,17]. 

There was a high risk in ten (59%) studies regarding the reference standard [13,15,16,19,20,21,22,24,27,28,29]. The flow and timing risk of bias was low for all the studies. In terms of applicability concerns, the participant selection bias was unclear in five (30%) studies, the index test bias was unclear in five (30%) studies and the reference standard was high in eight (47%) and unclear in four (24%) studies.

## 4. Discussion

This systematic review assessed biomarker expression before and after different types of treatment for peri-implantitis. This systematic review identified changes in 42 host-derived biomarkers. Several studies showed that samples from sites with peri-implantitis presented higher concentrations of inflammatory biomarkers compared to healthy sites [14,16,21,22,24,28].

This review shows that surgical treatment, such as flap debridement, alone can significantly reduce some investigated biomarkers (MMP-8 and TNF-a) in the long-term (12 months after surgery) [15,16], as well as significantly reducing clinical signs of inflammation (PD, BOP, suppuration, and PI). In terms of clinical parameters, surgical treatment with adjunctive antibiotics led to significant reduction of BOP, PD and suppuration at 3 months, compared to baseline. However, it is important to consider the biological cost-effectiveness of using adjunctive systemic antibiotics when considering this technique. Adjunctive use of EMD after debridement was reported to improve long-term implant survival. 

The clinical results showed that non-surgical treatment alone might minimally improve the clinical parameters and may lead to PD, BOP, PI and PICF volume reduction in the short-term (3 months) [16,21]. Also, non-surgical treatment results in minimal change in inflammatory biomarkers (IL-4, IL-10, and IL-12, TNF-a, RANKL, OPG IL-1β, IL-6, TNF-α, MCP-1/CCL2, MIP-1α/CCL3, IFN-γ, MMP-8, sRANKL, OPG and G-CSF levels). In addition, non-surgical treatment with adjunctive use of antimicrobials does not lead to complete disease resolution. However, significant improvement in terms of clinical parameters including PD, BOP and PI reduction were found with non-surgical treatment when combined with laser. 

This study reveals an association between peri-implantitis treatment and the stability of clinical outcomes. The present systematic review shows that favorable outcomes in term of clinical and biomarkers levels are correlated with surgical treatment of peri-implantitis. 

This systematic review identified changes in 42 host-derived biomarkers. Several studies showed that samples from sites with peri-implantitis presented higher concentration of inflammatory biomarkers compared to healthy sites [14,16,21,22,24,28].

### 4.1. Non-Surgical Treatment 

#### 4.1.1. Non-Surgical Treatment Alone 

During the treatment of peri-implant mucositis and peri-implantitis there were minimal effects of non-surgical treatment alone on the investigated inflammatory biomarkers (IL-4, IL-10, and IL-12, TNF-a, RANKL, OPG IL-1β, IL-6, TNF-α, MCP-1/CCL2, MIP-1α/CCL3, IFN-γ, MMP-8, sRANKL, OPG and G-CSF) [16,21]. The results showed that there is significant heterogeneity not only among the biomarkers that were investigated but also in the different clinical and laboratory methodologies that were implemented in the current studies. Therefore, a solid conclusion cannot be made regarding the efficacy of this treatment modality in modifying biomarker levels. The clinical results showed that non-surgical treatment alone might minimally improve the clinical parameters and may lead to reduction in PD, BOP, PI and PICF volume in the short-term (3 months) [16,21]. This would suggest that non-surgical treatment may create a more favourable conditions for surgical interventions in patients with peri-implantitis. 

#### 4.1.2. Non-Surgical Treatment with Adjunctive Use of Lasers

Adjunctive use of lasers may significantly reduce some of the investigated biomarkers such as IL-1b and IL-10 [13], but not other investigated biomarkers (IL-8, IL-1a, IL-6, TNF-a, MMP-1, MMP-3, MMP-8, MMP-9 and CRP) [13,23,26,28]. Improvement following non-surgical treatment with adjunctive use of lasers was shown to be significant in terms of clinical parameters including PD, BOP and PI reduction.

#### 4.1.3. Non-Surgical Treatment with Adjunctive Use of Antimicrobials 

Adjunctive use of AZM [18,22] or subgingival irrigation with chlorhexidine [24] were shown to have outcomes similar to non-surgical treatment alone, for both biomarker levels and clinical parameters. This is in accordance with other reviews investigating the clinical effectiveness of non-surgical techniques. It has been shown that these techniques can be effective in reducing clinical signs and symptoms of peri-implant inflammation (i.e., PD, BOP, and suppuration) but they have very limited potential to achieve complete resolution of the disease [31,32]. Moreover, adjunctive measures to non-surgical therapy are not effective in improving clinical parameters. The results of this systematic review have shown that professionally administered plaque removal techniques with adjunctive measures do not lead to complete disease resolution. Limitations and controversies in the literature might be due to heterogeneity in the case definition for peri-implant diseases (several studies were published before the release of the 2017 Consensus Classification) [3], variety in treatment approaches and methodology, difference in implant design, and variation in implant surface characteristics and defect configuration.

### 4.2. Surgical Intervention

#### 4.2.1. Access Flap Debridement Only

In peri-implantitis, access flap debridement alone can significantly reduce some investigated biomarkers (MMP-8 and TNF-a) in the long-term (12 months after surgery) [15,16]. Similarly, it can significantly reduce clinical signs of inflammation (PD, BOP, suppuration, and PI). It is well established that surgical intervention will lead to more attachment loss in periodontal and peri-implant tissues [15,16]. 

#### 4.2.2. Access Flap Debridement with Adjunctive Use of Lasers

In the present systematic review, only one study used adjunctive laser for treatment of peri-implantitis [28]. Participants were either smokers or non-smokers and were additionally subdivided into health/gingivitis group and periodontitis group. The concentration of MMP-8 levels following treatment significantly decreased 6 months post-surgical intervention in both smokers and non-smokers with periodontitis.

In this study [28], metronidazole 400 mg was prescribed for 10 days in 24 patients even though the exact number of patients that received antibiotics in each treatment group was not specified [28]. Due to limitations of this study [28], the clear benefit of adjunctive laser/PDT cannot be evaluated, and, thus, further studies are required to consider eventual molecular, biological, microbiological, and clinical effectiveness of this treatment modality.

#### 4.2.3. Adjunctive Use of Antibiotics to Access Flap Debridement 

In the current review, only one case-control study [14] used adjunctive prophylactic systemic amoxicillin/AZM in combination with access flap debridement surgery for the treatment of peri-implantitis and, in addition, assessed biomarkers (IL-1b, MMP-8, MIP-1a), and clinical parameters compared to healthy sites [14]. Although all biomarkers were reduced after treatment, the data showed only a significant decrease of MMP-8. In terms of clinical parameters, adjunctive antibiotics led to significant reduction of BOP, PD and suppuration at 3 months compared to baseline. 

An RCT [31] investigated the effectiveness of adjunctive antimicrobials in the surgical treatment of peri-implantitis and its relevant clinical efficacy has been reported at 12 months. Results showed that the adjunctive effect of systemic antibiotics is low and dependent upon implant surface characteristics [31]. It is important to consider the biological cost-effectiveness of using adjunctive systemic antibiotics when considering this technique.

#### 4.2.4. Regenerative Surgical Debridement with Enamel Matrix Derivative

In the present review, only one study used adjunctive EMD in combination with surgical debridement and it found a significant reduction in 40 PICF protein concentrations [17]. Unfortunately, no data regarding clinical parameters were reported in this study [17]. Adjunctive use of EMD was investigated in an RCT with 3–5 years of follow-up [32] and authors reported that adjunctive use of EMD improves long-term implant survival (100%) compared to control sites without adjunctive EMD (83%). As in previous studies, small sample sizes and differences in study designs do not allow the drawing of a solid conclusion sufficient to define a gold standard treatment of peri-implantitis.

## 5. Conclusions

In conclusion, inflammatory biomarkers were seen in increased concentrations in both peri-implant mucositis and peri-implantitis when compared to healthy peri-implant tissues. Based on the data from studies included in this systematic review, surgical treatment of peri-implantitis appears to be more effective in the resolution of inflammation as seen by changes in biomarker levels and clinical parameters in the short and long-term (up to 12 months). Surgical treatment in combination with EMD was found to be more effective in resolving inflammation up to twelve months.

Due to significant heterogeneity in the study designs, inclusion/exclusion criteria, biomarkers investigated, treatment protocols and the length of follow-up, these findings must be interpreted with caution. 

## Figures and Tables

**Figure 1 ijerph-19-14085-f001:**
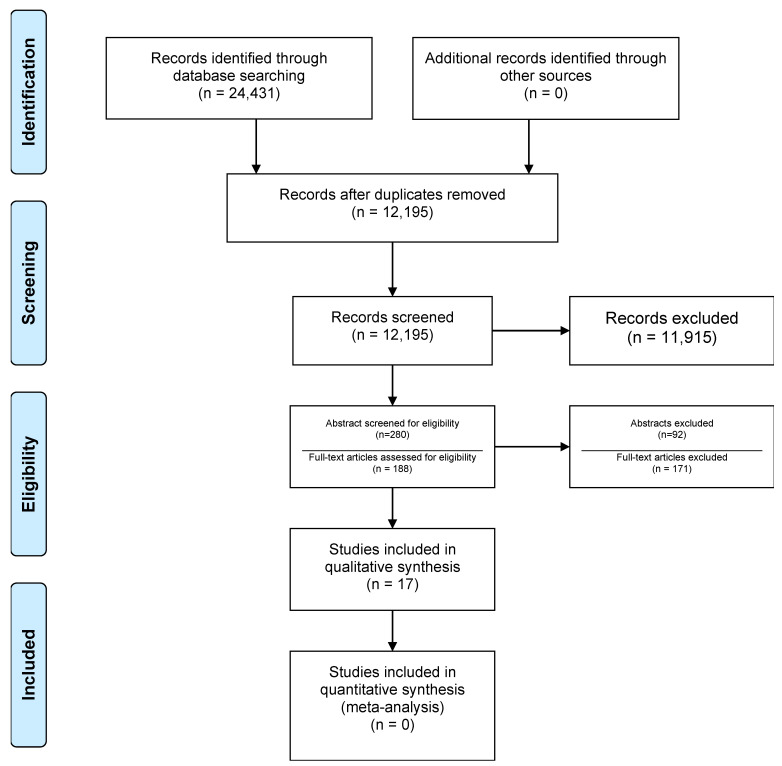
Flowchart of the study selection process (adapted from [9]).

**Table 1 ijerph-19-14085-t001:** Demographics and main characteristics of included studies. *A.a: Aggregatibacter actinomycetemcomitans*, AB: Antibiotic, ALP: Alkaline Phosphatase, AZM: Azithromycin, BI: Bleeding index, BOP: Bleeding on Probing, CAL: Clinical Attachment Loss, CCL- 5: Chemokine ligand- 5, CHX: Chlorhexidine, CRP: C- Reactive Protein, ELISA: enzyme-linked immunoassay, EMD: Enamel Matrix Derivative, F: Female, *F.n Fusobacterium nucleatum*, GI: Gingival index, GM-CSF: Granulocyte Macrophage Colony-Stimulating Factor, IFN-g: Interferon- gamma, IL-1b: Interleukin-1beta, LILT: Low Intensity Laser Treatment, LDD: Local Drug Delivery, M: Male, MBL: Marginal Bone Loss, MC: Minocycline Hydrochloride, MCP-1: Monocyte Chemotactic Protein-1, MIP: Macrophage Inflammatory Protein, MMP: Matrix Metallo Proteinase, MPO: Myeloperoxidase, MW: Mouthwash, n: number, NR: Not Reported, NSPT: Non-surgical periodontal treatment, NSSD: Not Statistically Significantly Different, OC: Osteocalcin, OFD: Open Flap Debridement, OPG: Osteoprotegerin, OPN: osteopontin, PCR: Polymerase Chain reaction, PDT: Photo Dynamic Therapy, PDGF: Platelet Derived Growth factor, *P.g: Porphyromonas gingivalis*, PI: Plaque Index, PICF: Peri Implant Crevicular Fluid, PIM: peri-Implant Mucositis, PIP:Peri- implantitis, PIH: Peri-implant Health, RANK: Receptor Activator of Nuclear factor Kappa-Β, RANKL: Receptor Activator of Nuclear factor Kappa-Β Ligand, RBL: Radiographic Bone Loss, rCAL: relative Clinical Attachment Loss, RCT: Randomised Clinical Trial, REC: Recession, SUP: Suppuration, *T.d Treponema denticola*, *T.f Tanerella forsythia*, Ti: titanium, TNF: Tumour Necrosis Factor, PPD: Probing Pocket Depth, TGF: Tissue Growth Factor, VEGF: Vascular Endothelial Growth Factor.

Investigator, Year (Country)	1. Methods 2. Study Type 3. Arm 4. Sample Size Calculation	Participants 1. Number; M/F 2. Distribution between Groups 3. Age (Mean) 4. Source of Recruitment 5.History of Periodontitis a. in total n (%) b. in Test Group n (%) c. in Control Group n (%)	Study Outcomes 1. Primary Outcome 2. Secondary Outcome 3. Other	1.Definition for Peri-implantitis 2. Implant Position 3. Type of Prosthesis	Type of Procedure/Clinical Details	Laboratory Technique
Bassetti et al., 2014 [13] (Switzerland)	1. RCT, 1 blind examiner 2. Parallel arm 3. 20 per group, power of 68%, standard deviation of 1.3	1. 40; 20 M/20 F 2. Equal Distribution between groups (20/20) 3. 58 4. Private practice and university 5.a. In total: 26 (65%) 5.b. In test group: 18 (45%) 5.c. In control Group: 8 (20%)	1. Number of BOP + sites 2. PPD, REC, CAL, microbiological (*Porphyromonas gingivalis*, *T. f*, *T. d*, *A. a*, *P.i*, *Campylobacter rectus*, *F. n*, *Capnocytophaga gingivalis*, *Parvimonas micra*, *Eubacterium nodatum*, *Eikenella corrodens*) and IL-1b, IL-8, IL-10, MMP-1 and MMP-8) change in PICF.	1. PPD of 4–6 mm with concomitant BOP at ≥1 peri-implant site and - Radiographic marginal bone loss ranging from 0.5 to 2 mm between delivery of the supra structure and pre-screening appointment 2. NR3.NR	Adjunctive LDD vs. adjunct PDT to Mechanical debridement with titanium curettes and a glycine-based powder air polishing for submucosal biofilm removal	1. Microbial analysis by Real-time PCR 2. Biomarker assessment by ELISA
Bhavsar et al., 2019 [14] (USA)	1. Case control 2. Study has 2 arms 3. NR	1. 48; 21/27 2. Sex-matched controls and cases 3. 66.37 4. University of Kentucky College of Dentistry 5.a. In total: 50% 5.b. In test group: 24 (100%) 5.c. In control Group: 0 (0%)	1. IL-1β, MMP-8, MIP-1α in PICF before and after surgical and anti- microbial therapy. 2. PPD, BOP, SUP, periodontal phenotype, implant mobility, plaque, amount of radiographic bone loss	1. PPD ≥ 4 mm and radiographic bone loss with more than 20%, but no more than half (50%) of the implant length when compared to baseline radiograph taken at least one year prior to baseline. 2. In health, 11 maxillary and 13 mandibular. In PIP, 7 maxillary and 17 mandibular	- Healthy implants vs. access flap surgery in PIP with anti- microbial therapy. - Prophylactic AB prior to surgery; Amoxicillin/azithromycin - Full thickness flap on buccal and palatal, debridement, 30 sec Tetracycline paste application - Post-operative AB for 7 days.	1. Biomarker selection using Luminex IS-100 instrument
De Mendonga et al., 2009 [15] (Brazil)	1. Case series 2. No arm 3. NR	1. 10; 5/5 2. 50/50 3. 62.3 ± 8.4 4. Guarulhos University 5. NR	1. Total amounts of TNF-a in the PICF 2. PI, mucosal marginal bleeding (MB), BOP, SUP, PPD, PD reduction, relative clinical attachment level (rCAL)	PPD ≥ 5 mm with BOP and/or SUP and concomitant radiographic bone loss involving at least three threads compared to the radiograph taken at the time of prostheses placement	Access flap surgery, debridement with abrasive sodium carbonate air-powder, and resin curettes CHX mw for 7 days.	Biomarker assessment with ELISA
Duarte et al., 2009 [16] (Brazil)	1. Case control 2. PIH, PIM and PIP 3. NR	1. 40; 15/20 2. 10 classified as PIH, 10 classified as PIM, and 20 classified as PIP 3. 53.4 ± 16.2 4. Guarulhos University	1.Visible plaque accumulation, marginal bleeding, BOP, SUP, and PPD 2. The total amounts of gene expression interleukin IL-4, IL-10, and IL-12, TNF-a, RANKL, and OPG in PICF	Presence of PD ≥ 5 mm with BOP and/or SUP and concomitant radiographic bone loss involving at least three threads of the implant but no more than half of the implant length.	PIM: mechanical debridement using abrasive sodium carbonate air-powder and resin curettes. PIP: open surgical debridement using abrasive sodium carbonate air-powder and resin curettes.	ELISA
Esberg et al., 2019 [17] (Sweden)	1. Pilot RCT 2. Surgical treatment of PIP without EMD 3. Pilot study-no sample size calculation.	1. 29 (4 drop out; at the end 25), gender not reported (not present nor original study). 2. 15 EMD group and 14 to the non-EMD group 3. No mean reported. The median age at implant installation was 70.0 years (min–max, 61–81) in the EMD group and 73.5 years (67–83) in the non-EMD group. 4. Not specified, all patients referred to periodontology department (Gävle County Hospital). 5. Most of the patients in this study had a history of periodontitis that had been successfully treated before peri-implant surgery, and 50% of the patients stated that tooth loss was due to periodontitis.	1. PICF proteome profile before and at 3, 6 and 12 months after the treatment of active peri-implantitis 2. PICF proteome profile relation with implant loss, BOP, PPD and adjunctive EMD treatment. smoking and implant loading time	1. PIP was defined as PD ≥ 5 mm with BOP and/or SUP and progressive angular peri-implant bone loss ≥ 3 mm as measured on radiographs 2. NR 3. Screw retained	Initial hygiene phase when needed. Access flap for mechanical cleaning using an ultrasonic cleaner with a special implant tip and titanium instruments combined with rinsing with sodium chloride solution (9 mg/mL, 2 × 20 mL). The randomization disclosed the allocation to either adjunctive EMD or no EMD at the implant site before closure of the flap.PICF Collection: Paper were placed for 30 sec at the implant mucosal sulcus site with the deepest pocket	Proteome: protein function and protein–protein interaction networks
Gershenfeld et al., 2018 [18] (Australia)	1. RCT 2. Study has one arm: AZM + NSPT vs. Placebo + NSPT 3. No SS calculation performed	1. 17; 8/9 2. Randomization 3. 61.9 (59.7 for test group and 64.4 for control group) 4. NR, Consecutive patients referred to the Westmead Centre for Oral Health (Westmead, Sydney, NSW, Australia) 5. In total 8 of total patients In test group: 5 In control group: 3	1. BOP and SUP, PPD and gingival margin retraction (recession in mm), radiographic bone loss 2. GI, PI and microbiological and IL-1 Beta results, AZM in PICF	1. Defined as having a PPD of 5 mm or more with BOP with or without SUP, and radiographic bone loss of more than 2 mm after abutment connection. 2. NR 3. NR	OHI and mechanical debridement for all pts.; test: AZM 500 mg once daily and control placebo for 3 days. Review at days 3, 7, 21, 90 and 180 after mechanical debridement for follow-up. PICF collected.	The total viable aerobic and anaerobic microbiological counts by conventional culturing methods. The AZM level in PICF was assayed using the Driscoll method (Carbonnelle et al., 2011). ELISA for IL-1b
Granfeldt et al., 2010 [19] (Norway)	1. Case control 2. No Arm 3. NR	1. 36; NR 2. NR 3. NR 4. NR	1. MMP-8 concentration before and 12 months after surgical treatment of PIP	1. NR 2. NR 3. NR	OFD alone vs. OFD + porous titanium particles	PICF sample analysis by ELISA
Hallstrom et al., 2016 [20] (Sweden)	1. Double-blind randomized placebo-controlled trial 2. One arm, effect of NSPT in PIM 3. Sample size calculation indicated that 23 patients in each group.	1. 46; 18/31 2. The study groups were slightly, but non-significantly, imbalanced with respect to general health and tobacco use at baseline; the proportion of smokers was higher in the test group, while the proportion of healthy patients was higher in the placebo group. The groups were, however, balanced regarding age and sex. 3. Placebo 63.3 years and test: 53.7 years 4. NR 5. NR	1. PPD 2. PI, BOP, SUP collected at baseline and after 1, 2, 4, 12 and 26 weeks, PICF (IL-1b, IL-1RA, IL-4, IL-6, IL-8, IL-17A, CCL5, TNF-a, IFN-g and GM- CSF) Plaque samples (*Porphyromonas gingivalis*, *Prevotella intermedia Prevotella nigrescens*, *Tannerella forsythia*, *A. actinomycetemcomitans*, *Fusobacterium nucleatum*, *Treponema.* *Denticola*, *Parvimonas micra*, *Campylobacter rectus*, *Porphymonas endodontis*, *Filifactor alocis*, *Prevotella tannerae*) Collected at baseline and after 1, 2, 4, 12 and 26 weeks.	1. PPD ≥ 4 mm combined with bleeding and/or pus on probing using a probing force of 0.2 N. 2. 31 maxillary implant (17 placebo and 14 test group) and 18 mandibular implants (8 placebo and 10 test group). 3. NR	After initial mechanical debridement and OHI, the patients received a topical oil application (active or placebo) followed by twice-daily intake of lozenges (active or placebo) for 3 months. The active products contained a mix of two strains of *Lactobacillus reuteri (probiotics)*	PICF Volume was recorded using a Perio- trone 8000.Biomarker concentration assessment by Bio-Plex Cytokine Assay.
Hentenaar et al., 2021 [21] (Netherlands)	1.Case control 2. 3 arms; healthy implant and PIP before and after treatment. 3. 40 implant (20/20) from 36 patients, average effect size of 0.9 and power of 80%.	1. 36; 22/14 2. Equal number of implant distribution between groups (20/20) 3. 60.2 4. University setting 5. NR	1. level of biomarkers IL-1β, IL-6, TNF-α, MCP-1/CCL2, MIP-1α/CCL3, IFN-γ, MMP-8, sRANKL, OPG and G-CSF in healthy implants and PIP before and 3 months after nonsurgical therapy. 2. PPD, BOP%, SOP%, Pi%, full mouth PPD (mm), full mouth SOP%, full mouth Bop%, full mouth PI%, MBL, Mean PICF Volume, mean periotron value.	1. Progressive loss of marginal bone ≥ 2 mm, as compared to baseline radiograph in combination with bleeding and or suppuration on probing. 2. NR 3. NR	PICF samples and clinical data recorded at baseline and 3 months after non-surgical treatment using Airflow Master Piezon (EMS)	Luminex assay to analyse biomarkers in the PICF
Kalos et al., 2015 [22](Australia)	1. Pilot prospective double-blind placebo controlled randomized clinical trial. 2. Study has 2 arms, NSPT and AZM vs. NSPT alone and compare to PIH 3. NR	1. 22 cases (17 PIP and 5 PIH) 9 in the test (4 M and 5 F) and 8 participants in the control (3 M and 5 F) 2. NSSD 3. For the test and control groups were 59.7 (SD = 13.1) and 64.4 (SD = 8.5) 4. Private practice 5. 8 patients in total, NSSD between groups	1. Mean counts and mean changes from baseline levels in the anaerobic and aerobic microbiological counts (CFU/mL) and the pro inflammatory cytokine Il-1β levels (pg/mL) over time. 2. AZM in PICF over time (expressed as a positive or negative result) To determine the frequency of “positive responders” to treatment. This was determined by the % frequency of patients who displayed a decrease in microbiological and immunological parameters from baseline.	1. Pocket probing depth of ≥5 mm with bleeding on probing with or without suppuration and radiographic bone loss of >2 mm after abutment connection 2. NR 3. NR	NSPT with and without AZM	aerobic vs. anaerobic, bacterial complexes by culture technique ELISA for IL-1 analysis Batch testing for presence or absence of AZM
Komatsu et al., 2018 [23](Japan)	1. RCT 2. Has 2 arms: Er:YAG and MC 3. 40 in total, 20 patients per group, effect size = 0.80, alpha = 0.05, and power at 80%)	1. 40 in total; 6/12 (laser) 9/10 (MC) 2.NR 3. 64.1 + −8.5 (laser) 64.8 + −7.2 (MC) 4. University setting 5. NR	1. Clinical parameter: PPD, CAL, BOP, Mo, and BL. IL-1α, IL-1β, IL-6, IL-8, TNF-α, and MMP-1, 3, 9, and 13, CRP 2. The count of G + and G-in both *groups (P. intermedia*, *P. gingivalis*, *T. forsythia*, *T. denticola*, *F. nucletum)*	1. PID: PPD greater than 5 mm and concomitant BOP from at least two sites. 2. NR 3. NR	Treatment (Er: YAG) irradiation Vs. locally delivered minocycline hydrochloride (MC)	PICF: biomarker detection by multiplex suspension array system, whereas C-reactive protein (CRP) Sub-gingival plaque: PCR
Malik et al., 2015 [24] (India)	1. Case control 2. Study has 2 arms 3. NR	1. 30 participants 20/10 2. NR 3. 41.83 ± 13.67 4. University setting 5. NR	1. PICF concentration of MPO and ALP 2. PI, GI, mPI, mBI, mGI, and PPD	1. Presence of supra- gingival plaque; mBI score >1, mPI score >1, mGI score >1; redness and swelling of peri-implant mucosa; radiographic evidence of bone loss higher than two-thirds length of the first step of implant or exposure of ≥2 threads of the implant; probing depth ≥4 mm in at least 1 site around the fixture. 2. NR 3. NR	Healthy implant Vs, NS anti-infective therapy (supra and sub gingival scaling and local irrigation with 0.2% CHX. Post-operative CHX gel for 4 weeks.)	ALP Liquid Stable Reagent Kit using modified DGKC method; MPO using spectrophotometric MPO assay.
Peres Pimentel et al., 2019 [25] (Brazil)	1. Double-blind, randomized, crossover study 2. Random assignment to Triclosan/fluoride (n: 13) or fluoride toothpaste (n: 13) 3. NR	1. 26; 15/11 2. Triclosan/Fluoride Toothpaste (n = 13) or Fluoride Toothpaste (n = 13) 3. 49.62 ± 16.01 years 4. University setting 5. Confounding factors: NR	1. Biomarker Levels of IFN-γ, IL-17, IL-1β, IL-10, IL-6, IL-8, TNF-α, OPG, osteocalcin (OC), osteopontin (OPN), MMP-2, MMP-96, TGF-β7, RANKL. 2. PI, BOP, PPM, PD and RCAL at experimental sites at baseline, 3-, 7-, 14- and 21-day follow-ups.	1. PIH: PDD < 4 mm with no Bop and no evidence of radiographic bone loss beyond bone remodelling (AAP, 2013) 2. NR 3. NR	All smoker patients 3 weeks not performing mechanical plaque removal and after 3 weeks randomly assigned to 2 groups: group 1: Triclosan fluoride toothpaste, group 2: only fluoride toothpaste 3DS. Clinical and biomarker assessment at days 0, 3, 7, 14, 21.	Biomarker assessment by MAGpixTM instrument (Luminex)
Renvert et al., 2017 [26] (Sweden)	1. Case series 2. No arm 3. NR	1. 41; NR 2. NR 3. NR 4. NR 5. NR	1. Pro- and anti-inflammatory cytokines IL-1β, IL-1ra, IL-6, IL-8, IL-17A, IP-10, MIP-1α, PDGFBB, TNF-α, and VEGF 2. Microbiome. *Aeruginosa*, *S. aureus and T. forsythia*. Clinical characteristics: BOP; sup, Bone level, PPD; at baseline and 6 months after treatment	1. Bone loss > 3 mm and probing pocket ≥ 5 mm, and with bleeding/pus on probing at the implant. 2. NR 3. NR	Nonsurgical therapy by either with the PerioFlow^®^ device or by an Er: YAG laser (KAVO, Biberach, Germany. Outcomes in stable (no further bone loss, probing pocket depth decrease ≥0.5 mm, no bleeding/suppuration) and unstable patients after 6 months.	Luminex magnet bead technology and checkerboard DNA-DNA hybridization
Ribeiro et al., 2018 [27] (Brazil)	1. Double-blind, randomized, crossover study No arm 2. NR	1. 22 Gender/Sex: 8/13 2. Triclosan (n = 11) or placebo (n = 11) 3. 48.45 ± 13.64 years 4. University setting 5. NR	1.Biomarker: IL-4, IL-17, IL-6, IL-23, INF-γ, TNF-α, MMP-2, MMP-9 2. Clinical evaluation of PI/%, BI, position of the peri-implant margin (PPM/mm): distance from the stent to the peri-implant margin, RCAL, PPD; at baseline and at 3, 7, 14, and 21 days	1. NR 2. NR 3. NR	3 weeks not performing mechanical plaque removal and after 3 weeks randomly assigned to 2 groups: group 1: Triclosan Fluoride toothpaste, group 2: Fluoride toothpaste	PICF analysis by MAGpix™ instrument10 (Luminex)
Thierbach et al., 2016 [28] (Germany)	1. case-control 2. PIP, PIH 3. NR	1. 29 patients; male n: 11 (healthy), male n: 14 20:8 (peri-implantitis) 2. Distribution between groups: age: 55.5 (PIH), 56.4 (PIP) 3. Age range: 4. German military hospital setting 5. 18 patients had periodontitis and 11 patients had healthy/gingivitis periodontium	1. PISF MMP-8 Levels in Peri-Implantitis 2. PPD, BOP, age of implant, smoking status	1. PPD > 5 mm in at least one site, exhibiting BOP and/or SUP, and having RBL in at least one site were considered implants with peri-implantitis. 2. NR 3. NR	Treatment: All patients underwent (aPDT) using a Low-Intensity Laser Treatment (LILT) laser of the implant pockets. The defects were then exposed to laser light with a wavelength of 660 nm for ten seconds using fibre optics. Light was delivered to six sites per implant. 4 months later the patients underwent access flap surgery of the PIP sites included in the study. The clinical treatment result was evaluated six months after the flap surgery.	ELISA for biomarker assessment
Wohlfahrt et al., 2014 [29](Norway)	1. Prospective, randomized, test-control, clinical study 2. Arm:2 3. NR	1. 32; NR 2. Distribution between groups: NR 3. NR 4. University and private 5. NR	1. MMP-8, IL-6, OPG, osteocalcin, leptin, osteopontin, parathyroid hormone, TNF-α, adiponectin and insulin, total protein content 2. PPD, BOP	1. NR1.NR 2. NR 3. NR	Surgery, comparing OFD and surface decontamination with titanium curettes and 24% EDTA gel (n = 16), or additional insertion of porous titanium granules.	MMP-8: ELISA, Luminex for the rest of biomarkers

## Data Availability

The protocol for this systematic review can be accessed at https://osf.io/jn97f/, accessed on 30 July 2022.

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
