# Peer review of "Biomarker Expression of Peri-Implantitis Lesions before and after Treatment: A Systematic Review"

_ijerph, 2022, doi:10.3390/ijerph192114085_

Round 1
Reviewer 1 Report
Introduction:
Please define the abbreviation when this is first mentioned (BOP) I suggest explaining in the introduction what are biomarkers and their functions in the periodontal tissues.
In this study most of the biomarkers studied were related to inflammatory response; could it be possible to explain it in the introduction.
For example, "the biomarkers present in the GCF and PICF are related to inflammatory response, like...."
I suggest not explaining the function of specific biomarkers into the introduction.
I recommend describing functions of biomarkers as an introduction and using the paragraphs of lines 70 to 88 into the discussion.
2.1 Study design
"The current systematic review...." please make a mention about figure 1. For example
"Followed the PRISMA.... (Figure 1)"
Lines 118 to 119, "The focus..." I did not understand the focus question. Authors evaluated the biomarkers expression related to peri-implantitis treated, or peri-implantitis or both? Please review.
1. Eligibility Criteria.
Adult patients were included in PICO. I consider is not necessary to describe in the eligibility criteria. Please review.
2. Selection of Studies
Results screening..., I consider the variables evaluated between reviewers; would it be used in this paragraph, for example: reviewers (HM, and AG.) ... "Taking in account, title screening, irrelevant studies, etc..."
3. results
Please do not repeat the kappa results; this was described in methodology.
"Biomarkers studied...."
This paragraph is hard to follow, mainly for the number of cites. Would it be possible to explain that is section was described in table 1, make a summary of this paragraph, and include the references in table 1?
For example
Biomarkers studied were interleukin, Macrophage Inflammatory Proteins (MIP), tumor necrosis factor (TNF) family, etc. Table 1 describes the biomarkers expressed on PICF and the methodology used (Laboratory technique)
4. Discussion
"This systematic review...."
I consider that the biomarkers were described in table 1, and it is not necessary to describe them again.
Overall comments
This manuscript is interesting, and authors made a great effort to do it, despite this manuscript is novel and interesting, the manuscript is repetitive sometime confuse between methods and results or results and discussion, authors repeat constantly the findings obtained and intermingle between methods, results, and discussion, mainly results and discussion. I suggest authors review the entire manuscript and make a better redaction avoiding intermingles between sections.
Reviewer 2 Report
Manuscript ID: ijerph-1869530
Title: Biomarker expression of peri-implantitis lesions before and after treatment: A systematic review
1.What is the main question addressed by the research?
To assess, through a systematic review of the literature, the changes in the expression of biomarkers in peri-implant crevicular fluid before and after treatment of peri-implantitis.
2.Is it relevant and interesting?
The article is relevant and interesting.
3.How original is the topic?
The topic is current.
4.What does it add to the subject area compared with other published material?
The authors have collected and analyzed original data.
5.Is the paper well written?
Yes, the article is well written.
6.Is the text clear and easy to read?
Moderate English editing is required.
7.Are the conclusions consistent with the evidence and arguments presented?
Yes, the conclusions consistent with the evidence and arguments presented.
8.Do they address the main question posed?
Yes, the Authors addressed the main question posed.
Other comments:
· English language: Moderate English editing is required.
· Introduction: This section needs few improvements. For example, Authors may include a brief sentence at the beginning of this section regarding innovations in implant dentistry based on the following reference: <<Innovative materials and technologies to improve treatment outcomes, reducing at the same time morbidity, biological, and surgical times are an intense research topic in implant dentistry [https://doi.org/10.3390/jpm12010108]>>.
· Materials and methods: Please follow PRISMA 2020 guidelines [https://doi.org/10.1136/bmj.n71]. Please better define the target of statistical analysis and please indicate the software used for analysis.
· Results: Please follow PRISMA 2020 guidelines [https://doi.org/10.1371/journal.pmed.1003583]. Please better define the results of statistical analysis. Figures quality is not so good. Please use other software (e.g. GraphPad)
· Discussion: What is the main theme that emerges from the authors' analysis? Is the type of peri-implantitis management a limitation? Please improve.
· Conclusion: This section has been properly prepared.
After making the indicated changes, I am available for a second round of peer review.
Thanks for the opportunity to review this manuscript.
Reviewer 3 Report
Dear Authors,
I have read with great interest your work entitled “Biomarker expression of peri-implantitis lesions before and after treatment: A systematic review.”
Here are some notes to improve the manuscript for publication, please provide a point-by-point response and highlight the modifications with a different color for each reviewer, so that it is easier to find them among the manuscript.
Abstract
-
You should summarize the results. In fact, the abstract is too long. Try to synthetize.
-
EMD acronym should be written in full the first time it is mentioned. The same should be done for the other acronyms (PD, BOP and so on). Moreover, emdogain is a registered trademark, it should be added the symbol in the text.
-
minocycline should not be written in in capital letter
Results
-
Figure 1 is too small. Please, enlarge it.
Supplementary materials: where they were uploaded? I could not find them in the mdpi area.
General notes
The article is too long. Long pieces of Results and Discussion sections should be summarized. I understand that this is a systematic review, however the article is very difficult to read and the Readers cannot focus on the main findings. If you are able, please try to reduce and rephrase.
English check is recommended.
Round 2
Reviewer 1 Report
Dear authors, Thank you for responding properly to my suggestions and resolving my question; the manuscript improved in relation to the previous.
Reviewer 2 Report
After the changes made the article may be suitable for publication.
Reviewer 3 Report
Dear Authors,
Thank you for providing the revised version of the manuscript.
The modifications performed make the manuscript suitable for publication.
Thank you for your hard work.